# Formulation of Protein-Rich Chocolate Chip Cookies Using Cricket (*Acheta domesticus*) Powder

**DOI:** 10.3390/foods11203275

**Published:** 2022-10-20

**Authors:** Ricardo S. Aleman, Jhunior Marcia, Shirin Kazemzadeh Pournaki, Isabel Borrás-Linares, Jesus Lozano-Sanchez, Ismael Montero Fernandez

**Affiliations:** 1School of Nutrition and Food Sciences, Agricultural Center, Louisiana State University, Baton Rouge, LA 70803, USA; 2Faculty of Technological Sciences, Universidad Nacional de Agricultura, Catacamas 16201, Olancho, Honduras; 3Department of Dairy and Food Science, South Dakota State University, Brookings, SD 57007, USA; 4Department of Analytical Chemistry, Faculty of Sciences, Avda Fuentenueva s/n, University of Granada, 18071 Granada, Spain; 5Department of Food Science and Nutrition, University of Granada, 18071 Granada, Spain; 6Department of Agricultural and Forestry Engineering, School of Agrarian Engineering, Universidad de Extremadura, 06007 Badajoz, Spain

**Keywords:** cricket protein, cookies, bioproduct, functional food, physical properties, rheology

## Abstract

In the Western world, the successful introduction of insect consumption may need awareness of insect ingredient benefits, and consumers’ anticipation of insect-based foods with sensory quality is crucial. The objective of this study was to develop protein-rich nutritional chocolate chip cookies (CCC) from cricket powder (CP) and analyze their physicochemical, liking, emotions, purchase intent, and sensory properties. The CP additions levels were 0%, 5%, 7.5%, and 10%. Chemical composition, physicochemical, and functional properties were analyzed using individual and mixed CP and wheat flour (WF). The proximate composition of CP mainly consisted of ash (3.9%), fat (13.4%), and protein (60.7%). In vitro protein digestibility of CP was 85.7%, whereas the essential amino acid score was 0.82. The CP inclusion significantly impacted the WF functional and rheological properties in all given incorporation levels in flour blends and doughs. The incorporation of CP produced darker and softer CCC, resulting from the effect of the CP protein. Adding 5% of CP did not impact the sensory attributes. Purchase intent and liking improved by using 5% of CP after panelists had revealed the beneficial information regarding CP. Concerning emotion terms, “happy” and “satisfied” significantly decreased while the negative emotion term “disgusted” increased among the highest CP substitute levels (7.5% and 10%) after beneficial information. Overall liking, flavor linking, education, consumption intent, gender, age, and positive emotion term “happy” were significantly assertive predictors affecting purchase intent.

## 1. Introduction

Human consumption of insects is known as entomophagy. Entomophagy is not a new trend in some parts of the world, including Africa and Southeast Asia, but it is still unfamiliar to many Western societies [1]. Few Western consumers seem willing to eat insects. The acceptance of entomophagy may depend on cultural aspects and emotional factors [2]. The reasons Westerners do not want to consume insects might vary from person to person. However, food neophobia, risk perception, sensory quality, and negative emotions are among the top aversions to trying insect foods. For example, negative emotions such as disgust and fear may cause Western consumers to avoid entomophagy [2]. Food neophobia refers to the fear of new or unfamiliar food, and in the Western world, the idea of insects as food is still unusual. Physical risks, such as safety concerns, can be perceived because consumers may not be able to distinguish between edible and disease-transmitting insects. Moreover, insect products might not meet consumers’ sensory expectations [3].

The United Nations expects the world population to increase by 2,000,000 people over the next 30 years and is expected to reach 1,100,000 by the end of the 21st century [4]. Due to the forecast population increase, there is a great concern about finding sustainable and innovative food solutions, such as using insect proteins for food purposes. [5]. Therefore, it is crucial to find alternative protein sources, as current livestock production practices are unsustainable. The introduction of insect meals as food ingredients to meet the demand for nutrients is gaining more and more success due to their significant nutritional contribution. Insect meals such as crickets, grasshoppers, and locusts have aroused interest as they have a protein content of over 50% as well as a high content of oleic, palmitic acids, and essential amino acids [6]. Studies carried out by Maiyo et al. 2022 used cricket flour for food use in the manufacture of porridge, with a high protein content (57%) and also has high total digestibility (88%), increasing the protein value in relation to the commercial value twice more [7]. On the other hand, authors such as Murugu et al. 2021 also studied the protein composition of cricket flour for its introduction into food products for species of crickets *S. icipe* and *G. bimaculatus*, obtaining protein percentages also higher than 50% [8]. In addition to presenting this high source of nutrients, cricket flour can be fermented with lactic acid bacteria, thus improving nutritional quality while also presenting a low impact on the consumption of environmental resources [9]. Another application of cricket flour for food purposes is the introduction of bread suitable for celiacs, providing the product with a high nutritional value and increasing its antioxidant properties [10]. The objective of this work was to elaborate on chip cookies, characterizing them nutritionally and sensory, and evaluating their acceptance among consumers.

## 2. Materials and Methods

### 2.1. Experimental Design

The UNAG Farms (Universidad Nacional de Agricultura, Olancho, Honduras) provided the cricket (Acheta domesticus) powder and wheat flour. Treatments consisted of three concentrations of cricket powder (0% (control), 5%, 7.5%, and 10%) singly added into chocolate chip cookie formulations made of wheat flour. Color, hardness, water activity, proximal analysis, sensory characteristics, liking, purchase intent, and emotions were evaluated in chocolate chip cookies. The rheology of the protein-rich nutritional chocolate chip cookies (CCC) dough was studied using a farinograph and alveograph. Furthermore, the cricket powder was analyzed to study protein digestibility and amino acid profile. The functional properties such as water absorption capacity, oil absorption capacity, emulsion capacity, foam capacity, foam stability, protein solubility, pasting properties, and particle size distribution of individual and mixed wheat flour and cricket powder were also studied.

### 2.2. Cricket Protein Digestibility and Amino Acid Profile

The experiments were performed in triplicates to study protein digestibility and amino acid profile. The protein digestibility of cricket powder was analyzed using the method of Tinus et al. (2012) [11]. Protein digestibility was studied by including 100 mg of cricket powder/wheat flour mixture (corrected on a weight basis for protein content of 65 mg protein) in 10 mL of distilled water. After stirring the solution for 1 h at 37 °C, the pH was adjusted to 8.0 using 0.1 M NaOH. Enzymatic digestion was applied using 13 mg of protease, 31 mg of chymotrypsin, and 16 mg of trypsin per 10 mL of distilled water. Enzymes are used together instead of separately to take advantage of the synergism between them. The solution was then adjusted to pH 8.0 using 0.1 M NaOH at 37 °C. A 1 mL aliquot obtained from an enzymatic cocktail was added to the sample solution, and the pH was measured every 30 s for 10 min. The protein digestibility was calculated using Equation (1):Protein digestibility (%) = 65.66 + 18.10 × pH10 min (1)
where pH 10 min is the differential of pH from the initial pH of 8.0 to the pH at the end of 10 min. For the amino acid profile, cricket powder was lyophilized and pulverized. The samples were submitted to protein hydrolysis by heating 25 mg aliquots with 0.5 mL of 6.0 N HCl for 24 h at 110 °C beneath vacuum conditions. Behind derivatization with o-phthalaldehyde (OPA) and 9-fluorenylmethyloxycarbonyl (FMOC), amino acid analysis was performed by reverse-phase high-performance liquid chromatography [12]. The score of essential amino acids was estimated as suggested by FAO/WHO (1991) [13].

### 2.3. Wheat Flour and Cricket Powder Functional Properties

Protein solubility was analyzed using 1 g of sample dissolved in 10 mL of distilled water in a test tube, and then the solution was placed in a vortex and stirred at 550 rpm for 30 min. The obtained solution was adjusted to pH 7 using 0.1 N NaOH. The suspension was centrifuged in a 10 mL test tube at 4500× *g* for 10 min. The supernatant was then collected into the sample for protein analysis in Kjeldahl digestion (AACC Method 46_13.01) [14]. Protein solubility was estimated by Equation (2):Protein solubility (%) = (Protein content of supernatant (%))/(protein content of sample (%)) × 100(2)

Similarly, the water absorption index (WAI) was studied by preparing the sample, as described above. The water absorption index (WAI) was calculated by Equation (3):WAI (%) = (Weight of supernatant (g))/(Dry Weight of cricket powder (g)) × 100(3)

For the oil absorption index determination, 1 g of cricket powder was added and mixed with 10 mL of soybean oil. The obtained mixture was then centrifuged in a 10 mL test tube at 1500*× g* for 10 min. The pelleted samples were removed and weighed after centrifugation, and the oil absorption index (OAI) was calculated by Equation *(*4):OAI (%) = (Weight of oil absorbed (g))/(Dry Weight of cricket powder (g)) ×100(4)

For foaming properties, the cricket powder solution (5% *w*/*w*) was adjusted to pH 7 and was stirred overnight at room temperature. The solution was then whipped using a mixer (KitchenAid Professional 5 qt Mixer, Ice Blue, Whirlpool, Benton Harbor, MI, USA) at the highest shear level. In a 100 mL graduated cylinder, the foam volume was evaluated at 1 min intervals from 0 min to 30 min. The foaming capacity and foam stability were determined using Equations (5) and (6), respectively:Foaming capacity (%) = (foam volume at 0 min (mL))/(Initial liquid volume (mL)) × 100(5)
Foaming stability (%) = (foam volume at 30 min(mL))/(foam volume at 30 min(mL)) × 100(6)

The emulsifying capacity of cricket powder was evaluated as the procedure of Yasumatsu et al. (1972) [15] with slight adjustments. For emulsion preparation, 7.5 g of cricket powder was emulsified in a 1:1 ratio of corn oil and 2.1 M NaCl using an immersion handheld blender (Waring Commercial, McConnellsburg, PA). The volume of the emulsified layer and the volume of the whole layer were measured using a mass cylinder. The emulsifying capacity was estimated using Equation (7):Emulsifysing capacity (%) = (volume of emulsified layer(mL))/(volume of whole layer (mL)) × 100(7)

The pasting properties of wheat flour and cricket powder were examined following AACC method 61.02.01 (AACC, 2009) [16] by using a Rapid Visco Analyzer (RVA) (RVA-4, Newport Scientific Pty. Ltd., Warriewood, Australia). The particle size of wheat flour and cricket powder was determined using a particle size analyzer (Microtrac S3500 series laser MicrotracBEL Corp, York, PA, USA) by wet measurement using isopropyl alcohol (IPA) as a mobile phase to evaluate particle size distribution from ~2- to 2000-µm.

### 2.4. Proximal Analysis of Flours and Chocolate Chip Cookies

The proximal analysis of wheat flour, cricket powder, and cricket chocolate chips was carried out by the Food Analysis Laboratory facilities of the Zamorano University, Valle del Yeguare, San Antonio de Oriente Municipality, Francisco Morazán Province, Honduras, including ash (AACC method 08_01.01), fat (AACC Method 30_20.01), moisture (AACC method 44_01.01), carbohydrate (differential method), and protein (AACC Method 46_13.01) [14].

### 2.5. Rheological Properties of Cricket Cookie Dough

The cricket cookie dough rheological determinations were measured in triplicates by alveograph, coupled with an alveolink integrator–recorder (Chopin Technologies, Villeneuve-La-Garenne, France) and with a farinograph (Brabender, Duisburg, Germany). The ISO 27971 (ISO 27971, 2008) methodology [17] was used over the alveograph test to determine the swelling index (SI), dough extensibility (DE), dough tenacity (DT), deformation energy (DEN), and curve ratio (DT/DE). The ICC 115/1 procedure [18] was used over the farinograph test to examine the degree of softening (DS), dough development time (DDT), water absorption (WA), and dough stability (ST).

### 2.6. Cricket Chocolate Chip Cookies Preparation

Ingredients used to make chocolate chip cookies containing 85% solids cricket powder (CCC) were wheat flour (300 mg/g), chocolate chips (200 mg/g), margarine (160 mg/g), sugar (130 mg/g), salt (25 mg/g), baking soda (4 mg/g), vanilla extract (6 mg/g), water (65 mg/g) and partial substitutions (0%, 5%, 7.5%, and 10%) of cricket protein. For chocolate chip cookies preparation, margarine and sugar were placed in the bowl in a stand mixer (KR876 Ankarsrum Mixer, Bread Beckers, Woodstock, GA, USA) and blended for 5 min using medium-high speed. Later, the baking soda, salt, whey cricket powder, and flour were put through a sieve and mixed at low speed for 5 min. The vanilla extract was mixed and added to the mixture, and the chocolate chips were combined and mixed at low speed for 2 min [19]. The dough was placed in the refrigerator for 2 h when it was developed and kneaded. After refrigeration, 12 g of dough was shaped and placed in a pan (Non-Stick Bakeware Muffin Pan & Cupcake Pan). The product was then heated and baked in the convection oven (Vulcan VC4GD Single Full Size Natural Gas Convection Oven, Sevier, TN, USA) at 180 °C for 9 min. The cookies were made with whey powder since the protein content in both powders does not show statistical differences. Hence, it is the closest thing to the test level.

### 2.7. Physicochemical Characteristics of Cricket Chocolate Chip Cookies

The color was analyzed using a colorimeter (BC-10, Konica Minolta, Inc., Osaka, Japan), including the L*, a*, b*, Chroma, Hue angle, and browning index values. The water activity (aw) of cricket chocolate chip cookies was measured using an aw meter (Hygrolab, Rotronic, Hauppauge, NY, USA). The hardness of the cookies was determined using a Texture Analyzer (model TA-XT2i, Stable Microsystems, Godalming, UK) through a 3-point bending test using a 3-point bending rig, trigger force of 30 g, and load cell of 45 kg. The textural properties were studied using a pretest speed of 1.6 mm/s, test speed of 2.2 mm/s, and post-test speed of 12 mm/s with a distance of 10 mm [20].

### 2.8. Consumer Study

This study was approved by the Honduran Association of Physicians-Nutritionists (ASOHMENU) with form # AS-ASHOMENU-005-2022. Emotional terms taken from the EsSense Profile*^®^* [21] were screened and considered for cookies using the check-all-that-apply (CATA) questionnaire (150 responses). Food-elicited emotions were chosen by >40% of respondents. Consumers (*n* = 180) from UNAG (Catacamas, Olancho, Honduras) participated in the sensory analysis using partitioned sensory booths. Panelists were given four chocolate chip cookie samples with a cup of water (Agua Azul, San Pedro, Honduras) and unsalted crackers (Nabisco, Northfield, IL, USA). The questionnaire was developed and made using Compusense*^®^* five software (Compusense Inc., Guelph, Canada), and all samples were enumerated using 3-digit random numbers with a counterbalanced design (t = 4, b = 4, k = 45). Panelists were introduced and instructed to evaluate color, aroma, flavor, texture, and overall linking (OL) on a 9-point hedonic scale. With tasting, purchase intent (PI) was examined on a “yes/no” scale, and emotions were analyzed using a 5-point intensity scale. Consumers were presented and introduced to the following beneficial information (HBI) statement(s): “This sample contains cricket powder, and it provides essential nutrients such as high content of protein (87.7% digestible) and sufficient essential amino acid and environmental benefits. According to statistics, poultry and pork require 385 and 340 megajoules of fossil fuel input per kilogram for protein output. On the other hand, cricket protein only requires 120 megajoules. Moreover, insects emit lower levels of greenhouse gases than cattle and use much less water than vertebrate livestock.” Once more, consumers evaluated overall liking, emotion intensities, and PI after receiving HBI.

### 2.9. Statistical Analysis

All statistical analyses were analyzed using SPSS 16 software (SPSS Inc., Chicago, IL, USA). A one-way analysis of variance (ANOVA) and post hoc Tukey test (α = 0.5) were used in sensory characteristics, emotional responses, and all physicochemical evaluations. The McNemar test was used to determine significant differences in PI before and after HBI. Multivariate analysis of variance (MANOVA) and descriptive discriminant analysis (DDA) was used to determine overall product differences across samples (considering sensory properties and emotional intensities simultaneously). A dependent *t*-test was used to determine differences in OL and emotion responses before and after the panelist acquired HBI. Logistic regression analysis (LRA) was conducted to analyze the effects of sensory characteristics and emotional terms on PI.

## 3. Results and Discussion

### 3.1. Cricket Protein Digestibility and Amino Acid Profile

The in vitro test indicated 87.7% protein digestibility (Table 1) for the cricket powder. These results are similar within the range of insect protein digestibility (77.9% to 98.9%)) [22,23,24]. Miech et al. (2017) [25], Norhidayah et al. (2016) [26], and Stone et al. (2019) [27] reported a digestibility of approximately 83%, 81%, and 76% for Cambodian field crickets and species of *Gryllus bimaculatus* and *Acheta domesticus*, respectively. The protein digestibility could be mainly affected by insect sex [27], insect fiber content [28], and processing treatments [29]. For the amino acid profile (Table 2), the cricket powder had high levels of glutamate and aspartate and low levels of cysteine and methionine (Table 2). Similarly, Poelaert et al. (2018) [30] and Stone et al. (2019) [27] have reported a similar trend. For essential amino acids, the cricket powder contained all essential amino acids, and all values for the essential amino acid score were more than one, indicating a surplus in essential amino acids (FAO/WHO 1991) [13]. Even though cricket protein could fulfill the minimum required essential amino acid content, insect protein is considered of less quality when compared to animal protein.

### 3.2. Functional Properties of Cricket Powder and Wheat Flour

The functional properties of individual WF and cricket powder are shown in Table 3. The water absorption capacity (137.5% and 175.2%), oil absorption capacity (143.7% and 140.4%), emulsion capacity (47.3% and 77.7%), foam capacity (74.4% and 15.9%), foam stability (1.9% and 80.5%), and protein solubility (20.3% and 30.5%) were found to be comparable to other studies [27,31]. For the blends of WF and CP, the incorporation of cricket powder in wheat flour significantly increased (*p* < 0.05) the water absorption capacity, foaming capacity, foaming stability, protein solubility, and emulsion capacity, while the oil absorption capacity was not affected. The increase in protein solubility with the addition of CP to WF may be associated with CP containing more albumins than WF. CP reported up to 3.15% of albumins (Stone et al., 2019) [27], while WF contains 2.01% [32]. Not surprisingly, the increase in water absorption capacity, emulsion capacity, foam capacity, and foam stability with the addition of CP to WF could be related to CP protein content (60.7%). Proteins are well known for their great binding water capacity [33], foaming properties [34], and emulsifying characteristics [35]. In our study, the oil absorption capacity of CP (137.5%) was identical to WF (143.7%). The oil absorption capacity of flours composite could be affected mainly by fat content and protein content in hydrophilic and hydrophobic groups [31].

Concerning pasting properties (Figure 1), the pasting temperature of all samples was between 87.55–89.93 °C. The addition of CP slightly increased the pasting temperature (PT). This phenomenon could be due to the larger particle size (Figure 2) and CP’s higher protein content (Table 1) when compared to WF. Bigger particles have a slower heat transfer rate and water diffusion [36]. Protein thermal conductivity is less when compared to starch, which is the main constituent of WF, leading to more time to generate viscosity during the heating cycle resulting in higher PT. The incorporation of cricket powder also resulted in an increase in final and setback viscosity and a decrease in peak viscosity and breakdown viscosity. Similarly, Khuenpet et al. (2020) [37] found an identical trend in insect powder such as mealworms (*Tenebrio molitor*). A high level of final and setback viscosity could lead to a high level of retrogradation [38].

### 3.3. Proximal Analysis of Cricket Powder, Wheat Flour, and Chocolate Chip Cookies

The proximal analysis of individual wheat flour, cricket powder, and chocolate chip cookies is presented in Table 1. The moisture (7.1% and 10.0%), ash (3.9% and 0.8%), protein (60.7 and 5.2%), fat (13.4% and 3.1%), and carbohydrates (7.7% and 80.7%) for CP and WF, respectively, were noticed within a similar content of other [27,39]. As recommended by the Code of Federal Regulation, both flours had a moisture content under the maximum limit (15%) (21 CFR 137.105) [40], and moisture content differences are commonly caused by the drying temperature and time applied in the milling process. Not surprisingly, cricket powder showed considerable fat, protein, and ash content. Commonly, edible insects are an excellent source of nutrients such as minerals providing B-complex, proteins, vitamins, and lipids with a lower omega-6 to omega-3 ratio [41]. The addition of cricket powder modified the nutritional composition of the chocolate chips cookies formulations. In the present formulation, the incorporation of cricket powder increased the moisture and protein content and decreased the ash, fat, and carbohydrate content. Currently, global trends in product development favor high protein and ash content and low fat and carbohydrate content [42].

### 3.4. Rheological Properties of Cricket Cookie Dough

The farinograph and alveograph test results of chocolate cookie dough are shown in Table 4. The addition of cricket powder showed a slightly increasing trend in water ab-sorption. This tendency could occur due to cricket protein content, as the protein’s hydro-philic groups have a water-binding capacity [33].

In insect protein, variations in water absorption could be due to the amino acid composition related to hydrophilic and hydrophobic groups [43]. When incorporating cricket flour into wheat flour, [39] reported a similar trend regarding the water absorption capacity of dough. As for dough development time (DDT), dough stability (DS), and dough tenacity (DT), the addition of cricket powder revealed an increase in DDT, DS, and DT. The reduction was minimal for DDT (2.7 to 3.2 min) and DS (7.5 to 8.8 min), whereas it was more meaningful for DT (81.7 to 99.3 mm H2O × 10^−4^). On the other hand, the inclusion of cricket powder showed a decrease in the degree of softening (DSS) (61.5 to 51.7 UB). Not surprisingly, the increase in DS, DDT, and DT and the decrease in DSS with cricket powder incorporation could be related to the cricket powder protein content. This phenomenon could be associated with the slowdown of gluten hydration and development caused by the interaction between cricket protein and wheat protein [44,45]. In other words, cricket protein could delay starch hydration and the development of the gluten network leading to an increase in DS, DDT, and DT and a decrease in DSS.

Concerning dough extensibility (DE) and index of swelling (SI), the addition of cricket powder showed a decline in the DE (87.5–77.1 mm) and SI (20.7–17.5 mm) values. DE and SI are related to gluten and starch content in flour blends [43]. The inclusion of cricket powder could possibly increase protein content and lower starch and gluten content. As for deformation energy (DEN) and curve ratio (DT/DE), these parameters increased with cricket powder incorporation at a minimum rate. This slight increase in DEN and DT/DE could be because of the differential between DT and DE on behalf of the increase in dough tenacity and decrease in dough extensibility. The DT/DE ratio has a critical role in the technological success of bakery products (Cappelli et al., 2019) [43]. In our results, the difference between treatments was minimal (0.91–1.3 DT/DE), having good changes in making cookies with cricket powder without causing substantial changes in the texture. When adding cricket flour to dough, [43] found a similar trend to our results regarding the farinograph and alveograph test results.

### 3.5. Physicochemical Properties of Chocolate Chip Cookies

The hardness of the CCC showed a marked effect on the cricket powder incorporation (Table 5). The CCC made with 5%, 7.5%, and 10% reported significantly (*p* < 0.05) lower hardness when compared to control samples. Another study corroborates that adding Mealworm powder (*Tenebrio molitor*) also lowers the hardness of crackers [46]. The hardness values could be associated with higher moisture content in CCC made with CP, meaning that spoilage bacteria do not have optimal conditions in this food matrix.

Our results from the proximal analysis confirmed this statement (Table 1). Similarly, the water absorption capacity of the dough and flour blend increased when CP was added (Table 3 and Table 4). Hardness plays an essential role in acceptability. For color, the L* values changed from slightly dark to a darker color (L* 65.34–54.95) when incorporating cricket. Generally, baked products get darker when free amino acid enhances Maillard reactions [47].

In our study, cricket protein most likely reacted with the carbohydrates present in the CCC formulation. Similarly, a* and b* values had the same pattern as L* values [48] and also reported a similar pattern regarding insect protein usage (*Tenebrio molitor*) in crackers. Significant differences in Chroma and ∆E values were found among the levels of cricket powder addition, suggesting that consumers can easily distinguish samples. For water activity, cricket powder inclusion did not affect the aw values. Commonly, protein could play a role in binding the water and reducing aw, but the amounts of cricket powder used in this study did not impact the aw values. The aw values of CCC were low (<0.17), meaning that spoilage bacteria do not have optimal conditions in this food matrix [48].

### 3.6. Sensory Characteristics of Chocolate Chip Cookies

Figure 3 shows the appearance of the biscuits with cricket proteins. For sensory properties, chocolate chip cookies had liking scores of ≥5.24 (Table 5). The CCC with 0% and 5% had the highest scores, whereas the CCC with 10% of CP reported the lowest scores. Texture and color liking scores had a similar pattern with OL. The CCC with 0% and 5% had the highest scores, and the CCC with 10% of CP reported the lowest scores. Compared with control CCC, the lower color liking scores for CCC with 10% were likely due to darker color (L* 65.34 and 54.95, Table 5) and a total color difference (∆E) value of 11.05–19.39, which is a perceptible contrast by human eyes (Hanmandlu, 2003) [49]. Another study also reports that color scores were negatively affected by cricket protein in crackers (Ardoni, 2021) [50].

Texture liking scores of all CCC varied from 6.19 to 6.05. Compared with the control CCC, the CCC with 10% of CP had a remarkably lower hardness (7721.07 vs. 1945.93 g force) (Table 2), which may partly contribute to a lower liking score for texture (6.49 vs. 6.05, Table 3). Some studies of insect powder incorporation have softer baked products, such as crackers, which were related to higher moisture content (Ardoni, 2021) [50]. Regarding overall aroma and flavor, liking scores ranged from (6.94 and 6.74, respectively) to (5.77 and 5.37, respectively) for all samples. The CCC with 0% and 5% had the highest scores, whereas the CCC made with 10% of CP reported the lowest scores. Commonly, flavor liking scores are strongly related to overall linking scores, and flavor perception must not contrast with familiar foods until the distinctive sensory qualities of insects are appreciated [3]. In the future development of chocolate chip cookies, flavor improvement should be the primary sensory quality to be addressed.

### 3.7. Effects of Beneficial Information on Overall Liking and Purchase Intent

The HBI has given to consumers significantly (*p* < 0.05) influenced OL scores only for CCC made, with 5% of CP giving a slight increase (from 6.05 to 6.59) (Table 5). There were no differences in overall liking scores among samples made with 7.5% and 10%. On the other hand, all samples were not significantly (*p* < 0.05) different in PI. A study [51] also reported no differences in purchase intent before and after HBI towards chocolate brownies containing edible-cricket protein. Usually, the beneficial claims are significant for initial liking [52]. Nevertheless, there is an influential dislike of insects as foods related to food neophobia and inadequately incentive to buy products made of insect ingredients, especially in the Western world [2].

### 3.8. Effects of Beneficial Information on Consumer Emotional Responses

The emotion scores of CCC made with different levels of CP are shown in Table 3. The scores of positive emotions (2.86–1.63 and 2.59–1.59, respectively, before and after HBI) were greater than the negative emotion scores (1.05–2.09 and 1.25–1.68) (Figure 4). However, revealing the presence of CP and its advantages also provoked negative emotional terms. For CCC made with 7.5% and 10% of CP, a significant (*p* < 0.05) decline for the positive emotions of “happy” and “satisfied” occurred when it was revealed the presence of CP, whereas “disgusted” significantly (*p* < 0.05) increased after tasting.

Gurdian et al. (2021) [51] also deduced that negative emotional terms such as “disgusted” and “worried” had a more significant influence on emotion profile after HBI on chocolate brownies containing edible-cricket protein, and a decrease has been observed for positive emotions. Typically, when HBI is given and revealed with beneficial claims to customers, positive emotional scores increase while negative emotions decrease [53]. Nevertheless, there is still a notable unwillingness to consume insects, especially in Western cultures, since entomophagy is still considered a disgust [54] and neophobia [55].

### 3.9. Overall Product Differences and Discriminating Sensory and Emotional Attributes

The Wilks Lambda test indicated that there were overall significant (*p* < 0.05) differences among all samples based on sensory characteristics and emotion terms (Table 6). The flavor reported the highest *p*-values (canonical correlation, cc = 0.669, 0.195, and 0.437 in the first, second, and third of their canonical, respectively), whereas color (canonical correlation, cc = 0.081, 0.122, and 0.019 in the first, second, and third of their canonical respectively) provided the lowest values among sensory properties to overall product differences. Insect flavors can differ depending on species and processing technique, and crickets had reported to provide a meat-like flavor [56].

These flavors could be unfavorable and evident at high levels in cookies. As a result, it suggested that flavor improvements are needed and required for future applications. The Wilks Lambda test showed that emotional responses did not have significant differences among all CCC. The emotional responses, including happy and satisfied, were the most dominant emotions that showed significant differences, specifically in the first canonical (Can 1). In other words, positive emotions were more suitable to show overall product differences than negative emotions [21]. Occasionally, OL does not correlate sufficiently with emotions, and products with low OL may evoke positive emotions and vice versa. However, OL and emotions jointly can satisfactorily illustrate consumption conduct [57].

### 3.10. Predicting Purchase Intent Using Logistic Regression Analysis

The CP incorporation in chocolate cookies was analyzed by ascertaining the effect of OL, consumption intent, education, gender, age, hedonic responses, and emotion terms on PI (before and after HBI) (Table 7). Gender, flavor, overall liking scores, and the emotional term “happy” were the most significant predictors for PI, including before and after getting the HBI. Similarly, another study has shown that taste, shape, and liking are the main drivers for purchase intent regarding brown rice cookies [58]. PI is related to how gender groups make purchase judgments [59], and the engagement with HBI is predominantly related to women [60]. The PI was not significantly affected by the beneficial information except for the 5% replacement of cricket protein, which reported an upsurge in PI after HBI. Consumers could be more aware of pricing and safety factors than the HBI concerning purchase intent.

## 4. Conclusions

It was accentuated by adding cricket (*Acheta domesticus*) powder to chocolate chip cookies to improve their nutritional profile. The influence of cricket protein on the sensory characteristics of chocolate chip cookies was more noticeable for flavor. As a result, flavor improvement is the proposed strategy when increasing the acceptability and purchase intent. The positive emotion scores were higher than the negative emotion scores. Nevertheless, positive emotion scores decreased while negative emotion scores increased after HBI was revealed and given to the customers. Cricket protein intrinsic characteristics should be adjusted or masked instead of showcased to improve familiarity. Preference mapping techniques and conjoint analysis should be used in future research for a better understanding of consumer behavior should be addressed.

## Figures and Tables

**Figure 1 foods-11-03275-f001:**
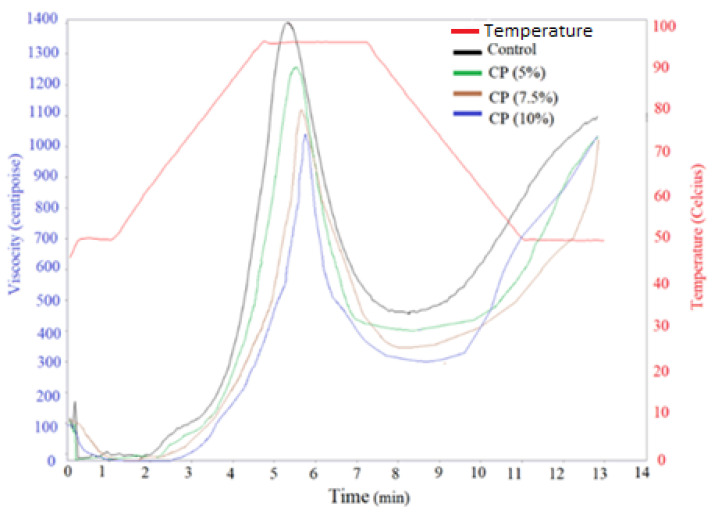
Viscosity profiles of wheat flour with cricket powder (CP) inclusion of 0%, 5%, 7.5%, and 10%, respectively.

**Figure 2 foods-11-03275-f002:**
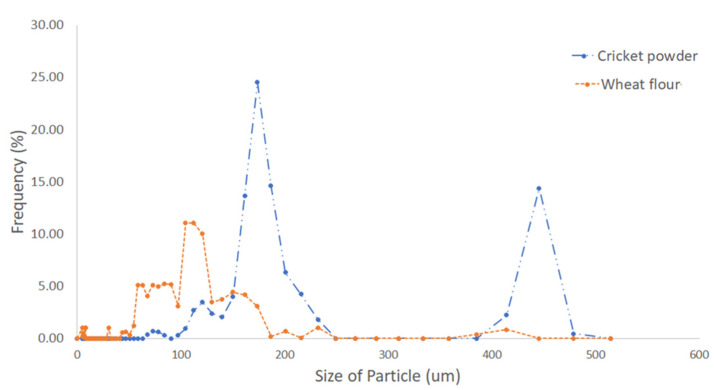
Particle size distribution of wheat flour and cricket powder.

**Figure 3 foods-11-03275-f003:**
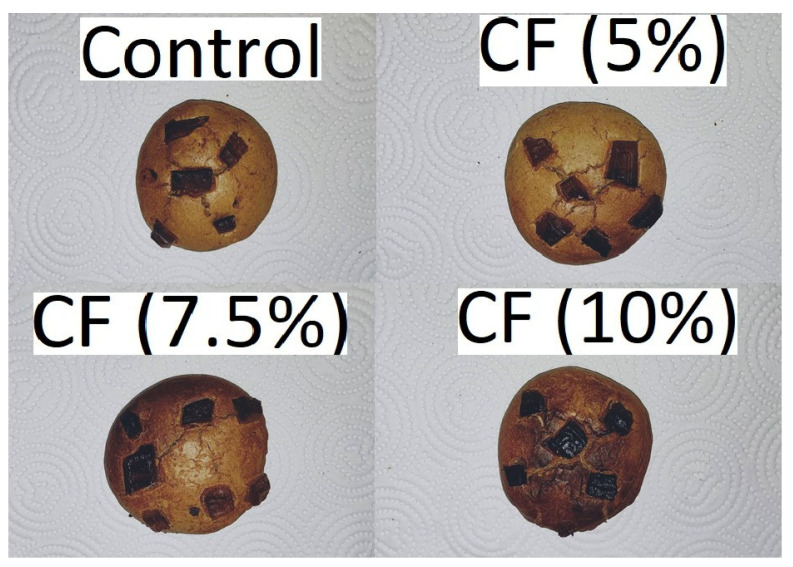
Appearance of the biscuits with cricket proteins.

**Figure 4 foods-11-03275-f004:**
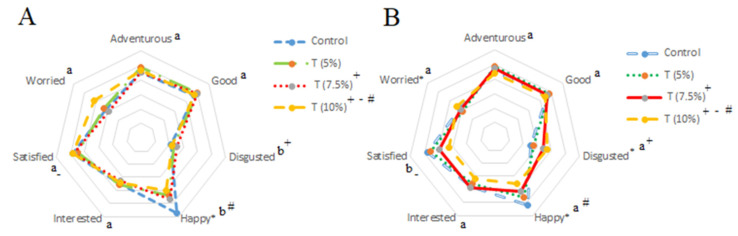
Mean emotion scores elicited by chocolate chip cookies with cricket powder treatments (T) inclusion of 0%, 5%, 7.5% and 10%, respectively. (**A**) Before consumers had been given information about cricket powder health and environmental benefits, and (**B**) after consumers had been given information about cricket powder health and environmental benefits. * Indicates significant difference among mean emotion scores by Tukey test (*p* < 0.05). + − # different letters indicate significant differences (*p* < 0.05) in emotion scores based on a dependent samples *t*-test, comparing before and after consumers had been given information about cricket powder health and environmental benefits.

**Table 1 foods-11-03275-t001:** Chemical composition of cricket powder.

Sample	Moisture (%)	Ash (%)	Protein (%)	Fat (%)	Carbohydrates (%)	In Vitro Protein Digestibility (%)
**CP^N/A^**	7.1 ± 2.74	3.9 ± 0.09	60.7 ± 5.28	13.4 ± 5.34	7.7 ± 1.35	87.7 ± 0.13
**WF^N/A^**	10.0 ± 2.57	0.8 ± 0.07	5.2 ± 2.11	3.1 ± 0.23	80.7 ± 1.35	Not measured
**CCC-CP 0%**	10.9 ± 0.54 ^c^	2.9 ± 0.03 ^a^	3.9 ± 0.45 ^d^	33.8 ± 1.34 ^a^	48.1 ± 1.33 ^a^	Not measured
**CCC-CP 5%**	12.1 ± 0.91 ^bc^	2.8 ± 0.02 ^ab^	8.9 ± 0.27 ^c^	31.5 ± 1.05 ^b^	44.4 ± 1.89 ^b^	Not measured
**CCC-CP 7.5%**	13.5 ± 0.82 ^b^	2.6 ± 0.03 ^bc^	11.9 ± 0.32 ^b^	30.2 ± 1.34 ^bc^	41.3 ± 1.33 ^c^	Not measured
**CCC-CP 10%**	15.1 ± 1.30 ^a^	2.5 ± 0.02 ^c^	13.3 ± 0.29 ^a^	28.5 ± 1.07 ^c^	39.8 ± 1.89 ^c^	Not measured

Legend: Cricket powder (**CP**), wheat flour (**WF**), and chocolate chip cookies (**CCC**). In vitro protein digestibility of cricket powder. Different lowercase letters mean a statistically significant difference between means followed (Tukey’s test *p* < 0.05 *n* = 3).

**Table 2 foods-11-03275-t002:** Amino acid (AA) composition and essential amino acid scores of cricket powder.

Amino Acid	AC **	CAS **
**Essential Amino acid** **His**	1.6 ± 0.12	1.3 ± 0.16
**Ile**	1.2 ± 0.07	1.6 ± 0.11
**Leu**	4.9 ± 0.05	1.2 ± 0.19
**Lys**	3.9± 0.07	1.1 ± 0.14
**Met ***	1.5 ± 0.11	1.4 ± 0.09 *
**Phe ^+^**	2.6 ± 0.12	1.0 ± 0.20 ^+^
**Thr ^+^**	0.7 ± 0.07	1.2 ± 0.20 ^+^
**Val**	3.5 ± 0.14	1.4 ± 0.21
**Non-Essential AA** **Ala**	5.1 ± 0.13	N/A
**Arg**	4.4 ± 0.10	N/A
**Asp**	5.7 ± 0.15	N/A
**Cys ***	0.7 ± 0.05	1.4 ± 0.09 *
**Glu**	7.5 ± 0.09	N/A
**Gly**	3.4 ± 0.05	N/A
**Pro**	4.1 ± 0.08	N/A
**Ser**	4.5 ± 0.04	N/A
**Tyr**	2.2 ± 0.06	N/A

Legend: abbreviation of amino acids. **Asp** = aspartate, **Tyr** = threonine, **Ser** = serine, **Glu** = glutamate, **Pro** = proline, **Gly** = glycine, **Ala** = alanine, **Cys** = cysteine, **Val** = valine, **Met** = methionine, **Ile** = isoleucine, **Leu** = leucine, **Tyr** = tyrosine, **Phe** phenylalanine, **His** histidine, **Lys** lysine, **Arg** arginine ** AC = amino acid composition and ** CAS = cricket powder essential amino acid score (0.82). The amino acid composition is expressed on a total amino acid weight basis except for tryptophan. The essential amino acid score is expressed on a total amino acid weight basis. * stands for the summation of MET and CYS. + stands for the summation of PHE and TYR.

**Table 3 foods-11-03275-t003:** Functional properties of wheat flour with cricket powder replacement levels.

Sample	WAC (%)	OAC (%)	FC (%)	FS (%)	PS (%)	EC (%)
**Control (0%)**	137.5 ± 11.32 ^a^	143.7 ± 7.61 ^a^	15.9 ± 4.74 ^c^	1.9 ± 0.04 ^d^	20.3 ± 0.32 ^a^	47.3 ± 3.11 ^c^
**CP (5%)**	144.3 ± 8.71 ^ab^	140.5 ± 6.83 ^a^	20.4 ± 5.34 ^b^	4.5 ± 0.07 ^c^	21.1 ± 0.54 ^a^	52.8 ± 5.34 ^bc^
**CP (7.5%)**	148.6 ± 9.54 ^bc^	144.7 ± 5.13 ^a^	22.8 ± 3.73 ^b^	7.1 ± 0.13 ^b^	21.8 ± 0.74 ^a^	55.2 ± 8.59 ^ab^
**CP (10%)**	155.9 ± 6.65 ^c^	137.5 ± 7.14 ^a^	24.7 ± 6.92 ^a^	10.7 ± 0.05 ^a^	23.5 ± 0.17 ^b^	60.5 ± 7.77 ^a^
**CP^N/A^ (100%)**	175.2 ± 5.87	140.4 ± 5.83	74.4 ± 9.36	80.5 ± 5.35	30.5 ± 0.94	77.7 ± 5.55

Legend: **WAC** = water absorption capacity, **OAC** = oil absorption capacity, **FC** = foaming capacity, **FS** = foaming stability, **PS** = protein solubility, **EC** = emulsifying capacity. Different lowercase letters mean a statistically significant difference between means followed (Tukey’s test *p* < 0.05 *n* = 3).

**Table 4 foods-11-03275-t004:** Farinograph and alveograph tests results of chocolate cookie dough.

Sample	* WA (%)	* DDT (Min)	* DS (Min)	* DSS (UB)	* DT (mmH2O) × 10^−4^	* DE (mm)	* SI (mm)	* DEN (J * 104)	* DT/DE (Ratio)
**Control (0%)**	58.5 ± 0.44 ^b^	2.7 ± 0.76 ^b^	7.5 ± 0.71 ^c^	61.5 ± 7.48 ^a^	81.7 ± 2.46 ^d^	87.5 ± 6.47 ^a^	20.7 ± 1.27 ^a^	255.3 ± 5.28 ^b^	0.91 ± 0.14 ^b^
**CP (5%)**	59.3 ± 0.5^8 b^	2.9 ± 0.42 ^b^	7.9 ± 1.02 ^b^	50.3 ± 5.01 ^b^	88.5 ± 2.90 ^c^	82.7 ± 2.46 ^a^	19.1 ± 1.07 ^a^	260.4 ± 4.69 ^ab^	1.0 ± 0.15 ^b^
**CP (7.5%)**	61.6 ± 0.65 ^a^	3.0 ± 0.70 ^ab^	8.5 ± 0.53 ^b^	50.0 ± 7.23 ^b^	91.3 ± 3.71 ^b^	79.9 ± 1.57 ^ab^	18.6 ± 1.05 ^ab^	260.9 ± 7.54 ^ab^	1.1 ± 0.13 ^ab^
**CP (10%)**	62.9 ± 0.54 ^a^	3.2 ± 0.37 ^a^	8.8 ± 0.34 ^a^	51.7 ± 2.96 ^b^	99.3 ± 6.35 ^a^	77.1 ± 5.73 ^b^	17.5 ± 1.14 ^b^	264.5 ± 5.31 ^a^	1.3 ± 0.16 ^a^

Means followed by different letters in the column differ by the Tukey test (*p* < 0.05). * The 0%, 5%, 7.5% and 10% CCC-CP treatments correspond to the CP addition of 0%, 5%, 7.5%, and 10%, respectively. SI = swelling index, DE = dough extensibility, DT = dough tenacity, DEN = deformation energy, DT/DE = curve ratio, DS = degree of softening, DDT = dough development time, WA = water absorption, ST = dough stability.

**Table 5 foods-11-03275-t005:** Physical–chemical properties and sensory characteristics of chocolate cookies with CP replacement levels.

Attribute	CP Replacement Levels			
	Control (0%)	T2 (5%)	T3 (7.5%)	T4 (10%)
**Physical–Chemical Properties**				
**L***	65.34 ± 1.2 ^a^	61.48 ± 1.4 ^b^	58.82 ± 1.5 ^c^	54.95 ± 1.7 ^d^
**a***	10.05 ± 0.6 ^a^	6.47 ± 0.5 ^b^	6.08 ± 0.2 ^b^	5.55 ± 0.4 ^c^
**b***	26.58 ± 0.5 ^a^	16.85 ± 1.4 ^b^	14.33 ± 1.1 ^b^	10.83 ± 2.3 ^c^
**Chroma**	28.41 ± 0.5 ^a^	18.05 ± 0.9 ^b^	15.56 ± 0.7 ^c^	12.17 ± 1.3 ^d^
**∆E**	N/A	11.05 ± 0.7 ^c^	14.43 ± 0.8 ^b^	19.39 ± 0.5 ^a^
**Water activity (Aw)**	0.15 ± 0.02 ^a^	0.16 ± 0.01 ^a^	0.13 ± 0.02 ^a^	0.14 ± 0.01 ^a^
**Hardness (g force)**	7721.07 ± 48.6 ^a^	5583.93 ± 32.9 ^b^	3591.75 ± 62.7 ^c^	1945 ± 20.1 ^d^
**Sensory Characteristics**				
**Color**	7.05 ± 1.5 ^a^	6.81 ± 1.4 ^ab^	6.41 ± 1.3 ^b^	5.95 ± 1.9 ^b^
**Aroma**	6.94 ± 1.6 ^a^	6.47 ± 1.5 ^ab^	6.34 ± 2.0 ^b^	5.77 ± 1.7 ^c^
**Flavor**	6.74 ± 1.5 ^a^	6.55 ± 1.7 ^b^	6.23 ± 1.2 ^c^	5.37 ± 2.1 ^d^
**Texture**	6.49 ± 1.8 ^a^	6.47 ± 1.3 ^a^	6.21 ± 1.5 ^ab^	6.05 ± 1.1 ^b^
**Overall liking**				
**Before**	6.23 ± 1.5 ^a^	6.05 ± 1.8 ^a^	5.77 ± 1.5 ^b^	5.24 ± 2.0 ^c^
**After**	6.75 ± 1.6 ^a^	6.59 ± 1.9 ^a+^	5.98 ± 1.4 ^b^	5.12 ± 1.6 ^c^
**Purchase intent** ( **%** )				
**Before**	62.75	58.50	52.75	46.50
**After**	61.50	65.25 +	55.00	43.50

Mean values in the same row followed by different letters are significantly different (*p* < 0.05) using the Tukey test. The cricket powder (CP) treatments correspond to the CP inclusion of 0%, 5%, 7.5%, and 10%, respectively. + Indicates significant differences (*p* < 0.05) in overall liking based on the dependent sample *t*-test and purchase intent based on McNemar’s test, comparing before and after consumers had been given the information about cricket powder health and environmental benefits.

**Table 6 foods-11-03275-t006:** Canonical structure describing group differences among low sodium roasted chicken.

Attribute		Can 1	Can 2	Can 3
**A**				
**Texture**		0.318	0.103	0.311
**Aroma**		0.254	0.255	0.207
**Flavor**		0.669	0.195	0.437
**Color**		0.081	0.122	0.019
**B**				
**Adventurous**	**Before**	−0.018	0.021	0.012
	**After**	0.127	0.012	0.125
**Good**	**Before**	−0.010	0.037	−0.002
	**After**	−0.485	0.117	0.025
**Satisfied**	**Before**	0.098	0.085	0.055
	**After**	0.186	0.028	0.109
**Worried**	**Before**	−0.054	−0.058	0.038
	**After**	−0.085	0.108	0.101
**Interested**	**Before**	0.003	0.048	−0.134
	**After**	−0.145	0.084	0.086
**Disgusted**	**Before**	−0.078	−0.134	0.094
	**After**	−0.046	0.154	0.132
**Happy**	**Before**	0.104	0.115	0.067
	**After**	0.126	0.117	−0.173
**Overall liking**	**Before**	0.614	0.069	0.157
	**After**	0.595	0.088	0.163
**Cumulative variance explained**		0.338	0.307	0.643
**MANOVA Wilks’ *p*-value**		0.029		

Based on the pooled within in-group variances. Can 1, 2, and 3 refer to the first, second, and third canonical discriminant functions, respectively.

**Table 7 foods-11-03275-t007:** Odds ratio for predicting purchase intent before and after beneficial information.

Attributes	Before		After	
	Pr > *X*^2^	Odds Ratio	Pr > *X*^2^	Odds Ratio
**Overall liking**	<0.001	1.294	<0.001	1.384
**Consumption intent**	<0.001	3.356	<0.001	2.956
**Gender**	0.036	1.576	0.027	1.745
**Age**	0.067	1.374	<0.001	0.574
**Education**	0.035	0.573	0.038	0.273
**Color**	0.463	1.743	N/A	N/A
**Flavor**	0.038	1.384	N/A	N/A
**Aroma**	0.086	0.694	N/A	N/A
**Texture**	0.183	1.956	N/A	N/A
* **Happy** *	0.047	1.856	N/A	N/A
**Interested**	0.184	1.783	0.081	1.447
**Good**	0.583	1.285	0.383	0.581
**Adventurous**	0.194	0.573	0.184	0.294
**Disgusted**	0.059	1.394	0.097	1.180
**Satisfied**	0.098	0.293	0.294	1.845
**Worried**	0.105	0.482	0.210	0.593

Analysis of maximum likelihood estimates was used to obtain parameter estimates. The significance of parameter estimates was based on the Wald *X*^2^ value at *p* < 0.05.

## Data Availability

Data is contained within the article.

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
