# Peer review of "Formulation of Protein-Rich Chocolate Chip Cookies Using Cricket (Acheta domesticus) Powder"

_foods, 2022, doi:10.3390/foods11203275_

Round 1

Reviewer 1 Report

The manuscript titled “Development of Protein-Rich Chocolate Chip Cookies using Cricket (Acheta domesticus) Powder” described the potential of using cricket powder and its significance in bakery industries. Authors determined various physicochemical characteristics of newly developed protein rich cookies. Although, I find the topic interesting and appreciate the authors' efforts, but many flaws need to be resolved.

Here are some remarks that have to be consider in further submission,

·         Introduction section- Author should clearly present, why they choose cricket powder. Authors have not mentioned a clear rationale for this study design, why they use cricket powder. It should be very clear with the recent references on what was lacking and what prompted the authors to conduct this study.

·         Material methods- AACC reference should be updated with recent one.

·         Results- Results presented in the form of table, figure is not properly placed, and legends are not very clear.

·         Discussion is missing, kindly add a separate section for discussion or see the Journal’s guidelines.

·         Authors are suggested to update some of the references, since some of references are very old one.

 Detailed review is been marked in pdf file, kindly refer to pdf pages of the manuscript

Author Response

Comments to Author

The manuscript titled “Development of Protein-Rich Chocolate Chip Cookies using Cricket (Acheta domesticus) Powder” described the potential of using cricket powder and its significance in bakery industries. Authors determined various physicochemical characteristics of newly developed protein rich cookies. Although, I find the topic interesting and appreciate the authors' efforts, but many flaws need to be resolved.

Here are some remarks that have to be consider in further submission,

Thanks for your appreciation. We have reviewed the manuscript point by point according to the reviewer.

Introduction section- Author should clearly present, why they choose cricket powder. Authors have not mentioned a clear rationale for this study design, why they use cricket powder. It should be very clear with the recent references on what was lacking and what prompted the authors to conduct this study.

Thank you very much. Updated references justifying the importance of the use of cricket flour in food have been introduced in the introduction.

Material methods- AACC reference should be updated with recent one.

Thank you very much. The last edition that exists of these methods of analysis is from the year 2000

Results- Results presented in the form of table, figure is not properly placed, and legends are not very clear.

Thank you very much. The figure has been placed correctly in the manuscript and the legends have been improved for clearer interpretation.

  • Discussion is missing, kindly add a separate section for discussion or see the Journal’s guidelines.

Thank very much. The discussion was improved with the introduction of new references.

  • Authors are suggested to update some of the references, since some of references are very old one.

Thank you very much. The old references were replaced by current ones, except in the analysis methods.

 Detailed review is been marked in pdf file, kindly refer to pdf pages of the manuscript.

Thank you very much. I have reviewed the PDF file with the corrections 

Reviewer 2 Report

Title: Suggesting the title be rephrased to "Formulation of...." (instead of "Development of..")

Introduction: Authors should give relevant examples to further justify their study (https://pubmed.ncbi.nlm.nih.gov/35407134/). Also, justify how the edible insect protein can contribute to meeting world food demand/ensure sustainable protein production.

Materials and methods: Why use a combination of 3 enzymes in protein digestibility and not use each enzyme separately?

Line 157, delete the word: Particle"

In CCC preparation, indicating the percentage CP is sufficient (line 186), so the description in lines 184-185 may be unnecessary.

Line 190 - 191, what is the essence of using two flavoring agents i.e., chocolate and vanilla?

Italize scientific names on line 251, and line 424.

Results

Line 337; Check table numbering for Table 3, Table 3 depicts results on the functional properties of WF and CP. For that matter, Table 1 indicates the results of the proximate analysis.

Table 2: Not mentioned in the text!

 The addition of product photos would add more value to this study!

 Conclusion:

Just conclude what was observed in the study without further justification.

Author Response

Comments to Author

  1. Title: Suggesting the title be rephrased to "Formulation of...." (instead of "Development of..")

Thanks a lot for the suggestion. It was changed in the title "Development of..." by "Development of".

  1. Introduction: Authors should give relevant examples to further justify their study (https://pubmed.ncbi.nlm.nih.gov/35407134/). Also, justify how the edible insect protein can contribute to meeting world food demand/ensure sustainable protein production.

Thank you very much, the following references were introduced: Maiyo et al., 2022; Murugu et al., 2021, Fombong et al., 2021, Vasilica et al., 2022 and Nissen et al., 2020.

  1. Materials and methods: Why use a combination of 3 enzymes in protein digestibility and not use each enzyme separately?

Thank you. Enzymes are used together instead of separately to take advantage of the synergism that exists between them. Also, as we are simulating what happens in the human body, these three enzymes are the ones that act in the stomach in greater proportion and always act together.

  1. Line 157, delete the word: Particle"

Thank you very much. The word particle was removed

In CCC preparation, indicating the percentage CP is sufficient (line 186), so the description in lines 184-185 may be unnecessary.

Line 190 - 191, what is the essence of using two flavoring agents i.e., chocolate and vanilla?

Thank you very much- Chocolate and vanilla are used as flavoring agents because according to Delgado et al., 2020, they are the standard ingredients used in chocolate chip formulations.

Italize scientific names on line 251, and line 424.

Thank you very much. Scientific names were placed in italics.

Results

Line 337; Check table numbering for Table 3, Table 3 depicts results on the functional properties of WF and CP. For that matter, Table 1 indicates the results of the proximate analysis.

Thank you very much. It was reviewed. Table 1 shows the proximal analysis of the flours used and the proximal analysis of the cookies. The other table presents the functional properties of the flours and the mixtures: Where it says table 3 we have made a mistake and it is Table 1.

Table 2: Not mentioned in the text!

Thank you very much. Table 2 was included in the text.

 The addition of product photos would add more value to this study!

 Thank you very much. A photo of the result of making the cookies was added to the text.

 Conclusion:

Just conclude what was observed in the study without further justification.

Thank you. This correction was made in the text.

Reviewer 3 Report

The authors have investigated Development of Protein-Rich Chocolate Chip Cookies using Cricket (Acheta domesticus) Powder. It is a new area of research that needs to be extensively studied using appropriate designs. The manuscript provides relevant information but needs restructuring and rewriting of some sections, particularly abstract and introduction. The relevance of entomophagy needs to be highlighted with scientific basis. The consumer study basis mentioned in study lacks objectivity. Can authors cite some references where such kind of studies have been undertaken? The results are presented well in tables but discussion part is very weak. Kindly provide references of similar studies and strengthen the findings through appropriate discussion. The conclusion mainly targets the emotional responses, however, the key findings seems to be missed which needs to be highlighted. In my opinion, the manuscript needs some improvement as mentioned above and can be considered after making necessary corrections.

Author Response

Comments to Author

The authors have investigated Development of Protein-Rich Chocolate Chip Cookies using Cricket (Acheta domesticus) Powder. It is a new area of research that needs to be extensively studied using appropriate designs. The manuscript provides relevant information but needs restructuring and rewriting of some sections, particularly abstract and introduction.

Thanks for your appreciation. We have reviewed the manuscript point by point according to the reviewer.

  1. The relevance of entomophagy needs to be highlighted with scientific basis. The consumer study basis mentioned in study lacks objectivity. Can authors cite some references where such kind of studies have been undertaken?

Thank you very much. Some references on this type of study were included in the introduction.

  1. The results are presented well in tables but discussion part is very weak. Kindly provide references of similar studies and strengthen the findings through appropriate discussion.

Thank you very much. The discussion has been improved by adding some references.

  1. The conclusion mainly targets the emotional responses, however, the key findings seems to be missed which needs to be highlighted. In my opinion, the manuscript needs some improvement as mentioned above and can be considered after making necessary corrections.

Thank you very much, the conclusion has been restructured.

Round 2

Reviewer 1 Report

The manuscript titled “ Formulation of f Protein-Rich Chocolate Chip Cookies using Cricket (Acheta domesticus) Powder” has been revise as per comment raised by reviewer. Although, I find that, there are some corrections still needed to be resolved.

Here are some remarks that have to be consider in further submission,

Line no - 26: rephrasing required, this statement is quite confusing.

Line no-  54-56: I didn’t find these data in reference 4, kindly check the data. Does the population by the end of 21st century will reduce compared to the coming 30 years ?

Line no - 74: Biological names- italic

Line no - 206:  Does the ingredient used in cookies preparation was pure cricket protein concentrate or cricket powder, kindly justify?

Line no - 209:  Does whey protein used as control ? What was the purpose of using whey proteins?

Line no - 206: Does the ingredient used in cookies preparation was pure cricket protein concentrate or cricket powder, kindly justify?

Table-1

·         Author mentioned proximate values in table 1 for cricket powder is 7.1, 3.9, 60.7, 13.4 and 7.7. However, total value of proximate composition should be around or equal to 100.  Kindly justify

·         What CP N/A    , WF N/A  stand for? Mention in table legends

Line no -264: Author should mentioned Result & Discussion

Line no - 384: Write short form for wheat flour etc, throughout the manuscript. Author should keep uniformity throughout the manuscript

Author Response

Reviewer 1

Comments to Author

The manuscript titled “ Formulation of f Protein-Rich Chocolate Chip Cookies using Cricket (Acheta domesticus) Powder” has been revise as per comment raised by reviewer. Although, I find that, there are some corrections still needed to be resolved.

Thanks for your appreciation. We have reviewed the manuscript point by point according to the reviewer.

Here are some remarks that have to be consider in further submission,

Line no - 26: rephrasing required, this statement is quite confusing.

Thank you very much. Indeed, that phrase was removed because it was confusing.

Line no- 54-56: I didn’t find these data in reference 4, kindly check the data. Does the population by the end of 21st century will reduce compared to the coming 30 years?

Thank you very much. I have modified the sentence because it was confusing.

Line no - 74: Biological names- italic

Thank you very much. They have been placed in italics.

Line no - 206:  Does the ingredient used in cookies preparation was pure cricket protein concentrate or cricket powder, kindly justify?

Thank you very much. It was 85% solids cricket powder.

Line no - 209:  Does whey protein used as control? What was the purpose of using whey proteins?

Thank you very much for your comment. The cookies were made with whey powder, since the protein content in both powders does not show statistical differences. So it is the closest thing to the test level. That is why it was used to develop the cookies as a control.

Table-1

  • Author mentioned proximate values in table 1 for cricket powder is 7.1, 3.9, 60.7, 13.4 and 7.7. However, total value of proximate composition should be around or equal to 100. Kindly justify

Thank you very much. Indeed, that column refers to the value of proteins, but if you add proteins plus carbohydrates, more lipids, more ashes, more moisture, it gives a percentage of 100%.

  • What CP N/A , WF N/A stand for? Mention in table legends

Thank you very much. The legend was introduced in table 1 with the abbreviations. Legend: Cricket powder (CP), wheat flour (WF) and chocolate chip cookies (CCC). In vitro protein digestibility of cricket powder. Different lowercase letters mean a statistically significant difference between means followed (Tukey’s test p < 0.05 n=3). Different capital letters mean significant differences between the treatments correspond (Tukey’s test p<0.05 n=3).

Line no -264: Author should mentioned Result & Discussion

Thank you very much. Results and discussion were included.

Line no - 384: Write short form for wheat flour etc, throughout the manuscript. Author should keep uniformity throughout the manuscript

Thank you very much. The acronyms were placed in the manuscript and standardized throughout the text.

Reviewer 3 Report

The authors have satisfactorily addressed all the points raised.

Author Response

Dear reviewer. Thank you very much for your input and contributions to improve the manuscript.
